# Knockdown of CLAUDIN-6 Inhibited Apoptosis and Induced Proliferation of Bovine Cumulus Cells

**DOI:** 10.3390/ijms232113222

**Published:** 2022-10-30

**Authors:** Wanjie Wang, Huiying Zou, Nanzhu Chen, Yaqing Tian, Haisheng Hao, Yunwei Pang, Xueming Zhao, Huabin Zhu, Dawei Yu, Weihua Du

**Affiliations:** Embryo Biotechnology and Reproduction Laboratory, Institute of Animal Sciences, Chinese Academy of Agricultural Sciences, Beijing 100193, China

**Keywords:** CLAUDIN-6, bovine cumulus cell, cell apoptosis, cell proliferation, cell cycle

## Abstract

This study aims to investigate the effects of CLAUDIN-6 (CLDN6) on cell apoptosis and proliferation of bovine cumulus cells (CCs). Immunofluorescence staining was used to localize CLDN6 protein in CCs. Three pairs of siRNA targeting CLDN6 and one pair of siRNA universal negative sequence as control were transfected into bovine CCs. Then, the effective siRNA was screened by real-time quantitative PCR (RT-qPCR) and Western blotting. The mRNA expression levels of apoptosis related genes (*CASPASE-3*, *BAX* and *BCL-2*) and proliferation related genes (*PCNA*, *CDC42* and *CCND2*) were evaluated by RT-qPCR in CCs with CLDN6 knockdown. Cell proliferation, apoptosis and cell cycle were detected by flow cytometry with CCK-8 staining, Annexin V-FITC staining and propidium iodide staining, respectively. Results showed that the *CLDN6* gene was expressed in bovine CCs and the protein was localized in cell membranes and cytoplasms. After CLDN6 was knocked down in CCs, the cell apoptosis rate significantly decreased and the pro-apoptotic genes *BAX* and *CASPASE-3* were down-regulated significantly, whereas the anti-apoptotic gene *BCL-2* was markedly up-regulated (*p* < 0.05). Additionally, CLDN6 knockdown significantly enhanced cell proliferation of CCs at 72 h after siRNA transfection. The mRNA levels of proliferation-related genes *PCNA*, *CCND2* and *CDC42* increased obviously in CCs with CLDN6 knockdown (*p* < 0.05). After CLDN6 was down-regulated, the percentage of CCs at S phase was significantly increased (*p* < 0.05). However, there was no remarkable difference in the percentages of cells at the G0/G1 phase and G2/M phase between CCs with or without CLDN6 knockdown (*p* > 0.05). Therefore, the expression of CLDN6 and its effects on cell proliferation, apoptosis and cell cycle of bovine CCs were first studied. CLDN6 low expression inhibited cell apoptosis, induced cell proliferation and cell cycle arrest of bovine CCs.

## 1. Introduction

Cumulus cells (CCs) are important somatic cells surrounding oocytes and a specific subtype of granulosa cells which provide nutrition for oocyte maturation as well as ovarian development [1,2]. CCs’ apoptosis can cause follicular atresia and their proliferation is closely related to follicular development [3,4]. This delicate balance has a crucial impact on ovarian development and oocyte maturation. Studies on the function and regulation of CCs’ growth can provide an important theoretical basis for ovarian development, oocyte maturation and subsequent embryogenesis.

Tight junctions regulate the transport of ions, macromolecules and proteins through the paracellular pathway and provide barrier function for the endothelium and epithelium [5]. The CLAUDIN (CLDN) protein family is a key tight-knit transmembrane protein and plays an important role in cell adhesion, polarity, permeability and glandular differentiation [6,7]. Currently, there are 27 members in the CLDN family that are distributed in different organs and play diverse roles. For instance, CLDN1 is mainly located in the heart, brain and lungs [8], CLDN4 is located in the ovaries, pancreas and salivary glands [9], and CLDN6 is located in embryonic stem cells, the early embryonic epithelium and fetal tissues such as the stomach, pancreas, lungs, breasts, kidneys, colon and ovaries [10,11]. CLDN6, one of the earliest proteins expressed in embryonic stem cells, is dedicated to the fate of epithelial cells [12]. As a specific surface marker of mouse and human pluripotent stem cells, CLDN6 significantly decreased its expression level during early differentiation, which was consistent with the down-regulation of pluripotency markers genes *Oct4*, *Sox2* and *Nanog* [13]. However, in epidermal supra-basal layers, CLDN6 overexpression resulted in a perturbed epidermal terminal differentiation program and hyper-proliferative squamous invaginations/cysts, replacing hair follicles and a lethal barrier dysfunction [14].

Additionally, CLDN6 protein was detected in mouse embryos at all stages before preimplantation. After activity of CLDN6 was inhibited, mouse embryos had no or an immature blastocoel cavity without expansion, and blastocyst formation was remarkably reduced [15]. In *Xenopus laevis*, knockdown of *cldn6* gene led to severe defects in pronephros tubular morphogenesis and blocked the terminal differentiation of tubule cells, which characterized by failure of apical accumulation of actin, reduced lateral expression of tight junction protein Na/K-ATPase [16]. 

Recently, the role of CLDN6 in cancer has attracted extensive attention from researchers as an ideal therapeutic target. Strong evidence indicates that CLDN6 is rarely expressed in healthy adult tissues. The altered expression of CLDN6 was linked to the development of various cancers whose malignant phenotypes include proliferation and apoptosis, migration and invasion and drug resistance [17]. In cervical cancer cells, the expression of CLDN6 was down-regulated and overexpression of CLDN6 suppressed cell proliferation, colony formation in vitro and tumor growth in vivo, which were accompanied and potentially caused by the promotion of tumor cell apoptosis [18]. However, in gastric cancer tissues and cell lines, the expression of CLDN6 protein was up-regulated and an increased CLDN6 level was associated with enhanced proliferation and invasion abilities of gastric cancer [19]. Similarly, CLDN6 was up-regulated in endometrial carcinoma compared with normal tissues and cell lines. Knockdown of CLDN6 significantly inhibited cell proliferation and the colony numbers of human endometrial adenocarcinoma cells via remarkable reduction of p-AKT, p-PI3K, and mTOR expression levels [20]. In summary, there is controversy in the different expression levels of CLDN6 in different cancers. However, it was confirmed that the variety of CLDN6 expression was associated with the occurrence of cancers.

At present, studies on the function of CLDN6 is mainly related to mice and humans, whereas those concerning livestock are rarely reported. Although CLDN6 takes part in cell apoptosis and proliferation in cancers, its regulatory role in bovine CCs is unclear. In this study, we first detected the mRNA and protein expression of CLDN6 in bovine CCs. Then cell apoptosis, proliferation, cell cycle and the expression levels of related genes were determined in bovine CCs with or without CLDN6 knockdown. Our finding will provide a theoretical basis to clarify the regulation effects of CLDN6 on cumulus cell growth in livestock.

## 2. Results

### 2.1. Expression of CLDN6 Gene and Protein in Bovine CCs

Isolated from bovine COCs matured for 22 h in vitro, cumulus cells were cultured in vitro. Under the microscope, monolayer CCs were round or oval in shape. The expression of the CLDN6 gene in bovine CCs cultured for 48 h was determined by real-time quantitative PCR (RT-qPCR) and immunofluorescence staining, respectively. Results showed that the CLDN6 gene was expressed in bovine CCs and the mRNA level was 1.5 times that of GAPDH (Figure 1). Its protein was distributed in cytoplasms and membranes (Figure 2).

### 2.2. Screening of Effective siRNA Sequences Targeting Bovine CLDN6 Gene

To select the most effective siRNA sequence targeting the *CLDN6* gene, the mRNA and protein levels of the *CLDN6* gene were examined by RT-qPCR and WB in bovine CCs cultured for 48 h after transfection of siRNA sequence targeting CLDN6 or siRNA universal negative control sequence (siNC, control group; Figure 3A,B). Results showed that all three pairs of siRNAs targeting CLDN6 significantly decreased the expression of *CLDN6* at mRNA and protein levels (*p* < 0.05, Figure 3C,D). Among those, the largest decline in *CLDN6* expression was found in bovine CCs transfected with siRNA-2 (*p* < 0.05). However, there was no significant difference in expression levels of *CLDN6* between CCs transfected with siRNA-1 and siRNA-3. Therefore, siRNA-2 was the most efficient sequence for *CLDN6* down-regulation and it could be used in subsequent experiments.

### 2.3. Effects of CLDN6 Knockdown on the Apoptosis of Bovine CCs

Apoptosis of bovine CCs transfected with siCLDN6 or siNC sequence (control group) were detected by flow cytometry. In Figure 4A, Q1, Q2, Q3 and Q4 refers to necrotic cells, late apoptotic cells, early apoptotic cells and viable cells, respectively. In the bovine CCs with CLDN6 knockdown, the cells’ apoptosis rate was significantly lower than that in cells of the control group (4.04 ± 0.68% vs. 7.74 ± 0.83%, *p* < 0.05, Figure 4B). The viable cells rate of CCs with CLDN6 knockdown was significantly higher than that in the control group (95.87 ± 0.76% vs. 92.17 ± 0.80%). The necrotic cells rates between two groups were at the same levels (0.10 ± 0.07% vs. 0.09 ± 0.03%, *p* > 0.05). For apoptosis-related genes, the expression of *BAX*, *CASPASE-3* and *BCL-2* were analyzed. As shown in Figure 4C, CLDN6 knockdown significantly down-regulated the mRNA levels of the pro-apoptotic genes *BAX* and *CASPASE-3* (*p* < 0.05). On the contrary, the anti-apoptotic gene *BCL-2* was significantly up-regulated in CCs with CLDN6 knockdown (*p* < 0.05).

### 2.4. Effects of CLDN6 Knockdown on the Proliferation of Bovine CCs

The proliferation of bovine CCs transfected with siCLDN6 or siNC sequence were detected at 24 h, 36 h, 48 h and 72 h after transfection with the CCK-8 kit. Results showed that the proliferation rates of CCs with or without CLDN6 knockdown showed the same trend, which decreased from 24 h to 48 h, then began to rise up to 72 h after siRNA transfection. Compared with the control group, CLDN6 knockdown significantly increased the CCs’ proliferation at 72 h after transfection (*p* < 0.05), whereas it had no influence at 24 h, 36 h and 48 h (*p* > 0.05, Figure 5A). Additionally, three proliferative genes, *PCNA*, *CCND2* and *CDC42,* were significantly up-regulated in CCs with CLDN6 knockdown (*p* < 0.05, Figure 5B–D).

### 2.5. Effects of CLDN6 Knockdown on the Cell Cycle of Bovine CCs

Bovine CCs transfected with siRNA or siNC sequence were stained with PI and the cell cycle was detected by flow cytometry (Figure 6A). According to Figure 6B, CLDN6 knockdown induced the significant accumulation of cells in the S phase. However, the cell proportion in the G0/G1 phase and G2/M phase showed no significant difference in CCs with or without CLDN6 knockdown (*p* > 0.05).

## 3. Discussion

CLDN6 belongs to a tight junction family and regulates cell proliferation, differentiation, invasion, metastasis, apoptosis and other processes through a variety of factors [21]. Previous studies found that CLDN6 was not expressed in healthy adult tissues but was expressed in various cancers and many types of embryonic epithelia [10,17]. Furthermore, CLDN6 was used as a surface marker of mouse pluripotent stem cells and an ideal therapeutic target in cancer [13,17]. However, little is known about the expression and function of CLDN6 in livestock adult cells. In the present study, the distribution and expression of CLDN6 in bovine CCs were first determined. Furthermore, CLDN6 knockdown had effects on apoptosis and the proliferation and cell cycle of bovine CCs.

Generally, CLDN6 is a tetraspanin transmembrane protein located within the tight junctional complex [22], in the cell membrane of primary tumors, gastric cancer cells and mouse stem cells [23,24]. In mouse neonatal kidneys, CLDN6 was highly expressed and distributed on the membrane of proximal renal tubules, thick ascending ramus, distal convoluted tubules and collecting tubules [10]. However, increasing evidence showed that beside the membrane, CLDN6 was weakly expressed in the cytoplasm of different cells such as breast cancer tissues and adjacent tissues and atypical teratoid/rhabdoid tumors [25,26,27]. The delocalization of CLDN proteins from cell membranes was common in transformed cells and ovarian cancer, which is associated with tumor cell migration and invasion [28]. Additionally, overexpression of CLDN6 with cytoplasmic tail domain deleted mis-localized CLDN6, CLDN10, CLDN11 and CLDN18 to the cytoplasm, which emphasized the importance of the CLDN tail domain in membrane targeting [29]. In addition to those mentioned, the healthy heart of a 14-year-old person showed diffuse cytoplasmic staining of cardiomyocytes and smooth muscle surrounding the vessels, without membrane staining [30]. Similarly, the CLDN6 protein was observed to be localized in the cytoplasm and membrane of normal bovine CCs by immunofluorescence staining in this study. CCs surround oocytes in COCs and regulate oocyte maturation through communication between them. Similarly, oocytes tightly control CCs to complete functions for oocyte development. Whether the reason for CLDN6’s appearance in cytoplasms of bovine CCs is related to oocyte requirements needs further research.

It is well known that the balance between cell apoptosis and proliferation is crucial to maintain the normal physiological state of multicellular organisms. Cell apoptosis is regulated by the pro-apoptotic protein BAX and anti-apoptotic protein BCL-2 with its key executioner CASPASE-3. In our study, the expression levels of *BAX* and *CASPASE-3* were significantly down-regulated, whereas *BCL-2* was significantly up-regulated in bovine CCs with CLDN6 knockdown. Furthermore, CLDN6 knockdown relieved cell apoptosis and promoted proliferation of bovine CCs, which was consistent with previous studies in human breast epithelium cell lines and cervical carcinoma [31,32]. However, CLDN6 acts as a tumor promoter to increase cell proliferation of human hepatocellular carcinoma cells and gastric cancer [19,33]. CLDN6 knockdown significantly inhibited endometrial cancer cell proliferation via the PI3K/Akt/mTOR signaling pathway [20]. In the human breast cancer cell MCF-7, for clones transfected with *CLDN6*, the level of apoptosis signal-regulating kinase 1 (ASK1) protein and mRNA was up-regulated, which indicates that the ASK1 signal may be associated with the pro-apoptosis effect of CLDN6 [34]. In summary, CLDN6 possesses pro- or anti-cancer effects in different cancers through its regulation of cell proliferation and apoptosis via different pathways. However, in the present study, the way in which CLDN6 modulated bovine cumulus cell growth was not involved. Whether the way is similar to those above or unique in cumulus cells is interesting since CLDN6 is lowly or not expressed in most adult tissues.

Cell proliferation is an important life activity of organisms, which is the basis of growth, development, reproduction and inheritance. According to our results, cell proliferation rates increased significantly in bovine CCs 72 h after transfection with the siRNA targeting *CLDN6* gene, which was are consistent with previous studies in cervical cancer cells [18], whereas it was contrary with the results in gastric cancer cell lines and human endometrial adenocarcinoma cells [19,20]. For proliferation-related genes, CCND2 is the rate-limiting factor of cell mitosis from the G1 phase to S phase and positively accelerates proliferation during folliculogenesis [35]. It was expressed in the GCs of ovaries and its mRNA levels were promoted by FSH. Female mice with CCND2 deficiency were sterile because GCs could not proliferate and arrested the pre-antral follicle growth [36]. CDC42 is a kind of Rho-GTPase and regulates cell cycle, apoptosis and proliferation [37]. The up-regulated CDC42 inhibited cell apoptosis and promoted proliferation of porcine and chicken GCs [38,39], whereas mice lacking CDC42 in podocytes developed congenital nephropathy and died from renal failure within 2 weeks after birth [40]. CDC42 was required for tight junction formation and knockdown of CDC42 impaired junction formation in bronchial epithelial cells [41]. PCNA participated in a variety of processes of DNA metabolism, including DNA replication and repair, chromatin organization and transcription and sister chromatid cohesion [42]. PCNA underwent dynamic fluctuation during cell cycles, with a peak at the S phase, and then it was used as marker of cell proliferation [43]. In both oocytes and somatic cells during the development of fetal and neonatal mouse ovaries, the expression pattern of PCNA was closely correlated with the meiotic prophase I progression when primordial follicles started to assemble. Knockdown of PCNA in cultured mouse ovaries strikingly increased primordial follicle assembly, but contained fewer granulosa cells [44]. However, the effects of CLDN6 on the expression of *PCNA*, *CCND2* and *CDC42* genes was not estimated in other studies. In this study, CLDN6 knockdown increased the mRNA level of the above three genes in bovine CCs.

Simultaneously, CLDN6 knockdown induced cell cycle arrest in bovine CCs and cells accumulated in the S phase. In human MCF-7 cells, a high expression of CLDN6 blocked cells at the G0/G1 phase [45]. Furthermore, in breast cancer cell lines (MDA-MB-231 cells), miR-7 and miR-218 resulted in an increase of cells at the G1 phase and a decrease of cells at the S phase with reactivation of CLDN6 by means of epigenetic switches in DNA methylation and histone modification [46]. However, in gastric cancer, in cells such as AGS, MKN7 and NUGC-3, CLDN6 silencing contributes to cell cycle arrest at the G1-S checkpoint through increasing p21^WAF1/Cip1^ and p27^Kip1^ and decreasing S-phase kinase-associated protein 2 (SKP2) protein levels, which are well-known cell cycle regulators [26]. Obviously, the cell phase arrested by the variety of CLDN6 levels was distinct in different kinds of cancer.

## 4. Materials and Methods

All experimental protocols were conducted in accordance with the requirements of the Institutional Animal Care and Use Committee of the Chinese Academy of Agricultural Sciences. All the chemicals used in this study were purchased from Sigma Chemicals Co. (St. Louis, MO, USA), unless otherwise indicated.

### 4.1. Culture of Bovine CCs

Bovine CCs were separated from the cumulus-oocyte complexes (COCs) matured in vitro. According to the procedure described previously [47], ovaries were collected from a local slaughterhouse and transported back to the laboratory within 2–3 h after washing with normal saline containing 100 IU/mL penicillin and 100 mg/mL streptomycin sulfate (1% PS) at 30 °C. COCs were aspirated from 2–6 mm follicles from clear ovaries. Only COCs with a homogeneous cytoplasm and intact cumulus cell layer were picked up for maturation in vitro in 4-well dishes (Nunc, Roskilde, Denmark) with 750 μL of M199 medium (Gibco, Life Technology, Carlsbad, CA, USA) containing 10% fetal bovine serum (FBS; Gibco, Life Technology, Carlsbad, CA, USA), 0.01 IU/mL follicle-stimulating hormone (FSH), 1 IU/mL luteinizing hormone (LH) and 1 µg/mL estradiol (E2) at 38.5 °C in an atmosphere of 5% CO2 and 100% humidity.

COCs were digested with 0.1% hyaluronidase for 3 min after they were matured for 22–24 h. M199 solution containing 10% FBS was used to stop digestion. The cell suspension was recovered after centrifugation twice at 1000 rpm for 5 min with DPBS (Gibco, Life Technology, Carlsbad, CA, USA) containing 1% PS. The cells of 1 × 10^5^/mL were cultured in a 6-well plate (Corning, Corning, NY, USA) with DMEM/Han’s F12 (DF12, Gibco, Life Technology, Carlsbad, CA, USA) medium containing 10% FBS and 1% PS at 37 °C, 5% CO2 and 100% humidity. After culturing for 48 h, CCs were trypsinized and seeded in the 6-well plate format at a density of 1 × 10^5^/mL.

### 4.2. Synthesis and Transfection of Small Interfering RNA (siRNA)

Based on the sequence of the bovine *CLDN6* gene (Accession No.: NC_037352.1) provided by the NCBI database, three pairs of siRNA targeting CLDN6 together with one pair of siRNA universal negative control sequence (siNC) were designed and synthesized by GenePharma Co., Ltd. (Shanghai, China). The siRNA sequences are listed in Table 1. According to the protocol described previously [48], four pairs of siRNA (100 nM) were conducted into bovine CCs at 60–70% confluency with Lipofectamine 3000 (Gibco, Life Technology, Carlsbad, CA, USA), respectively. Briefly, 3 µL of siRNA and 3 µL of Lipofectamine 3000 were respectively diluted with 100 µL serum-free medium Opti MEM (Gibco, Life Technology, Carlsbad, CA, USA) for 5 min followed by the mixing of the two solutions for 20 min. CCs were cultured with 1.5 mL serum-free Opti-MEM medium after they were washed twice with DPBS. The mixture containing Lipofectamine 3000 and siRNA was added into wells and incubated for 6 h. Then, cells were cultured with DF12 containing 10% FBS and 1% PS. The transfection efficiency was observed under fluorescence microscope (Nikon, Tokyo, Japan), and the cells were collected for subsequent experiments 48 h after transfection. The CCs transfected with siNC were the control group.

### 4.3. Real-Time Quantitative Polymerase Chain Reaction (RT-qPCR)

The gene expression levels between CCs with or without CLDN6 knockdown were analyzed with the CFX96TM system (Bio-RAD, Hercules, CA, USA). The CCs transfected with siRNA against bovine *CLDN6* gene or siNC grew for 48 h and were harvested for RT-qPCR. A total of eight genes were selected, including bovine CLDN6 gene, apoptosis-related genes *BAX*, *BCL-2* and *CASPASE-3*, proliferating cell nuclear antigen gene (*PCNA*), cyclin D2 gene (*CCND2*), cell division cycle protein 42 gene (*CDC42*) and reference gene *GAPDH*. Total RNA was isolated from CCs with RNeasy Micro Kit (Qiagen, Hilden, Germany) according to the manufacturer’s instructions. cDNA was synthesized using M-MLV reverse transcriptase (Promega, Madison, WI, USA). RT-qPCR volume was 15 µL, including 0.5 µL of upstream and downstream primers, 2 µL of cDNA template, 7.5 µL of TB Green Premix Ex Taq II (2×) and 4.5 µL of RNase free DD H2O. The RT-qPCR program steps were 95 °C for 30 s, 39 cycles of 95 °C for 5 s, 60 °C for 30 s. Each sample was replicated three times and data were normalized to the expression level of *GAPDH* in each sample. The 2–∆∆Ct value, where ∆Ct = Ct _gene_ − Ct _GAPDH_ and ∆∆Ct = ∆Ct _siRNA_ − ∆Ct _control_, was calculated to represent the relative mRNA expression levels of genes in the control and interference groups. Primer sequences are listed in Table 2.

### 4.4. Immunofluorescence Staining

CLDN6 protein in the bovine CCs with or without CLDN6 knockdown was localized using immunofluorescence staining. Cells of 1 × 106/mL were seeded directly on cover glasses (Solarbio, Beijing, China) in a six-well plate. When the cells reached 60–80% confluency, the cover glasses were removed from the medium and washed with DPBS three times. Subsequently, the cells were fixed with 4% paraformaldehyde for 1 h and then permeated with 0.5% Triton X-100 for 40 min at room temperature. After being washed three times with DPBS containing 1% BSA (PBS-1% BSA) for 10 min, cells were incubated in PBS-1%BSA overnight at 4 °C to block the unspecific binding of antibody. Primary antibodies against CLDN6 (1:500, Merck Millipore, Darmstadt, Germany) were incubated with cells overnight at 4 °C followed by Alexa Fluor 488 goat anti-rabbit IgG (1:1000, Cell Signaling Technology, Danvers, MA, USA) at 37 °C for 1 h. Finally, 5 mg/mL DAPI (Beyotime, Shanghai, China) was used to stain cell nuclei for 5 min and the cover glasses were placed on the slides. Photographs were taken under a fluorescence confocal microscope (Leica, Wetzlar, Germany).

### 4.5. Western Blotting (WB)

The bulleted lists is as follows: The expression of CLDN6 protein in the bovine CCs with or without CLDN6 knockdown was detected by WB. Forty-eight hours after siRNA transfection, CCs reached 80–90% confluency and about 5 × 10^6^ cells were collected. The total protein was extracted after cells were treated with lysis buffer. Protein concentration was quantified with a BCA kit (Beyotime, Shanghai, China). Protein samples were resolved by SDS-PAGE (12% acrylamide gel containing 0.1% SDS) and transferred by electrophoresis to a nitrate cellulose membrane (BioTraceNT, Pall, Port Washington, NY, USA). After blocking with 5% skim milk for 1 h, the membranes were incubated with rabbit antibody against CLDN6 (1:500; Merck Millipore, Darmstadt, Germany) or mouse antibody against β-TUBULIN (1:2000; Proteintech Group, Rosemont, IL USA) overnight at 4 °C. Goat anti-rabbit IgG-HRP (1:5000; Gene-protein Link, Beijing, China) and goat anti-rat IgG-HRP (1:5000; Gene-protein Link, Beijing, China) were incubated as secondary antibodies at 37 °C for 1 h. Enhanced chemiluminescence (ECL) reagent (Mei5 Biotechnology, Beijing, China) was used to detect protein. Exposed in a chemiluminescence instrument, the images of protein bands were quantified with ImageJ software (1.8.0, NIH, Bethesda, ML, USA).

### 4.6. Apoptosis Analysis

Apoptosis rates of the bovine CCs with or without CLDN6 knockdown were detected using an Annexin V-FITC (AVF)/propidium iodide (PI) double-staining apoptosis kit (Beyotime, Shanghai, China). The CCs transfected with siRNA or siNC were cultured for 48 h, digested with 0.25% trypsin and collected. After being centrifuged twice at 1000 rpm for 5 min with DPBS, cells were stained for 10 min with 5 µL AVF and 50 µg/mL PI at room temperature. Cells at a density of 1 × 10^6^/mL were analyzed within 1 h by FACS Verse flow cytometry (Becton Dickinson, Franklin Lakes, NJ, USA). Viable cells were AVF and PI negative, cells in early apoptosis were AVF positive and PI negative, cells in late apoptosis were AVF and PI positive, and necrotic cells were AVF negative and PI positive. The rates of apoptotic cells were presented as the sum of early and late apoptosis rates.

### 4.7. Cell Proliferation Assay

A cell counting kit-8 (CCK-8, Dojindo, Kumamoto, Japan) was used to detect the proliferation of bovine CCs with or without CLDN6 knockdown. According to the manual, WST-8 is a tetrazolium salt and functions as the chromogenic redox indicator. It can form an orange color and water-soluble formazan upon reduction by cellular dehydrogenases. The intensity of the color is proportional to the number of living cells. Bovine CCs were digested with 0.25% trypsin and seeded at a density of 2 × 10^3^ cells/well in 96-well plates (Corning, Corning, NY, USA). After 24 h, cells were transfected with siRNA against the *CLDN6* gene or siNC. Then, 10 μL CCK-8 solution was added to each well at 24 h, 36 h, 48 h and 72 h after transfection and retained for 1 h at 37 °C without light. The absorbance was detected at 450 nm by an enzyme labeling instrument (Tecan, Zürich, Switzerland).

### 4.8. Cell Cycle Analysis

A cell cycle and apoptosis analysis kit (Beyotime, Shanghai, China) was used to detect the percentage of CCs at different phases of cell cycle. CCs transfected with siRNA against *CLDN6* gene or siNC were cultured for 48 h and were harvested as described in apoptosis analysis. After twice washing with DPBS, cells were incubated with precooled 70% ethanol overnight at 4 °C. Following removal of 70% ethanol by centrifugation, cells were stained with PI for 30 min at 37 °C. DNA content was detected by Flow cytometry at 488 nm. The data were analyzed by Flow Jo (V10, TreeStar, Ashland, OR, USA) and the percentage of cells in the G1, S, M and G2 phase was calculated.

### 4.9. Statistical Analysis

All experiments were repeated at least three times. Data in each experiment are expressed as the mean ±SD. Statistical analyses were performed using one-way ANOVA with Duncan’s test for post hoc analysis using SAS version 9 (SAS Institute Inc., Cary, NC, USA). *p* < 0.05 was considered statistically significant.

## 5. Conclusions

This study observed for the first time the expression of CLDN6 in bovine cumulus cells cultured in vitro. Furthermore, CLDN6 knockdown significantly inhibited apoptosis of bovine CCs and induced S phase arrest and cell proliferation. These findings provide a reference for revealing the function and regulation mechanism of CLDN6 in cumulus cell growth and oocyte and embryo development of livestock.

## Figures and Tables

**Figure 1 ijms-23-13222-f001:**
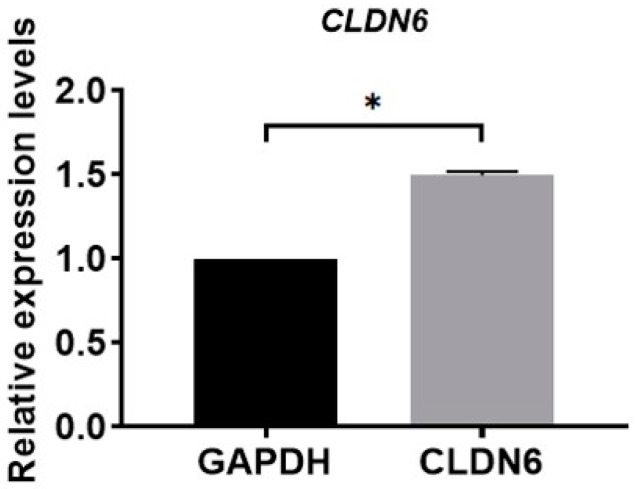
Relative mRNA level of *CLDN6* gene in bovine CCs was analyzed using RT-qPCR. Results were normalized to the mRNA level of *GAPDH*. Data are shown as means ±SD from three independent experiments. * *p* < 0.05.

**Figure 2 ijms-23-13222-f002:**
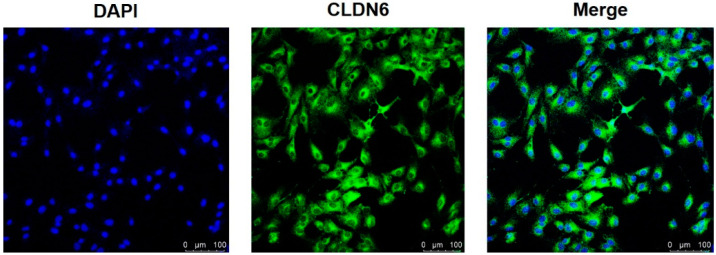
CLDN6 protein in bovine CCs was localized by immunofluorescence staining. The nuclei of CCs were stained with DAPI and viewed under ultraviolet fluorescence. CLDN6 was labeled with Alexa Fluor 488 and viewed under blue fluorescence. Bar = 100 µm.

**Figure 3 ijms-23-13222-f003:**
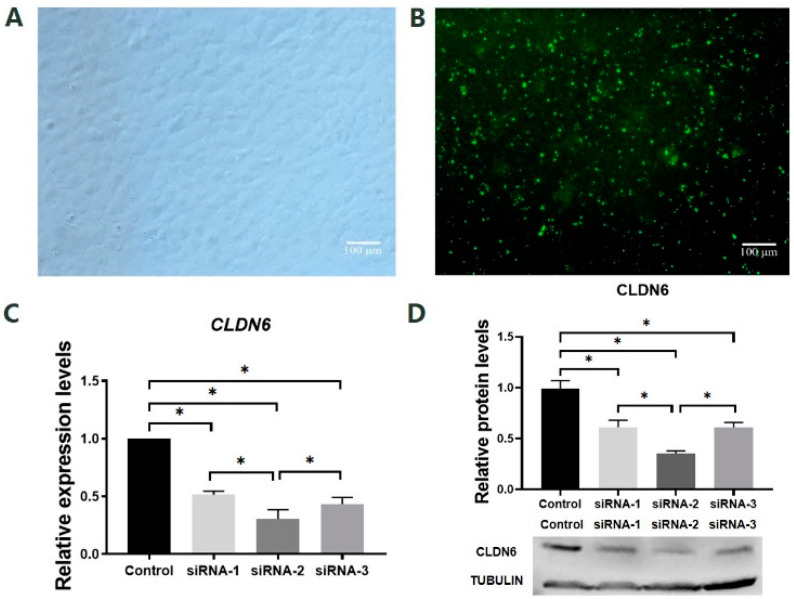
Effects of siRNA targeting *CLDN6* gene on its expression at mRNA and protein levels in bovine CCs. (**A**) CCs before siRNA transfection; (**B**) CCs transfected with siRNA; (**C**) The mRNA levels of the *CLDN6* gene in cells transfected with different siRNAs; (**D**) CLDN6 protein levels in cells transfected with different siRNAs. Control group was cells transfected with siNC. * *p* < 0.05. Bar = 100 μm.

**Figure 4 ijms-23-13222-f004:**
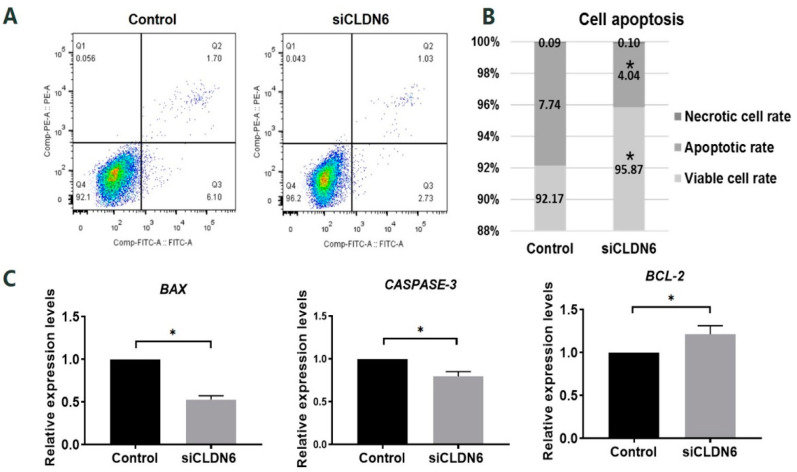
Apoptosis assay of bovine CCs with CLDN6 knockdown. (**A**) CCs transfected with siCLDN6 or siNC were stained with Annexin V-FITC and PI. The percentages of apoptotic cells were detected by flow cytometry. Q1 refers to necrotic cells, Q2 refers to late apoptotic cells, Q3 refers to early apoptotic cells, Q4 refers to viable cells; (**B**) Differences in apoptosis rate, viable cells rate and necrotic cells rate of CCs transfected with siCLDN6 or siNC were shown; (**C**) The mRNA levels of *BAX*, *CASPASE-3* and *BCL-2* genes in CCs transfected with siCLDN6 or siNC. Control group was the cells transfected with siNC. * *p* < 0.05.

**Figure 5 ijms-23-13222-f005:**
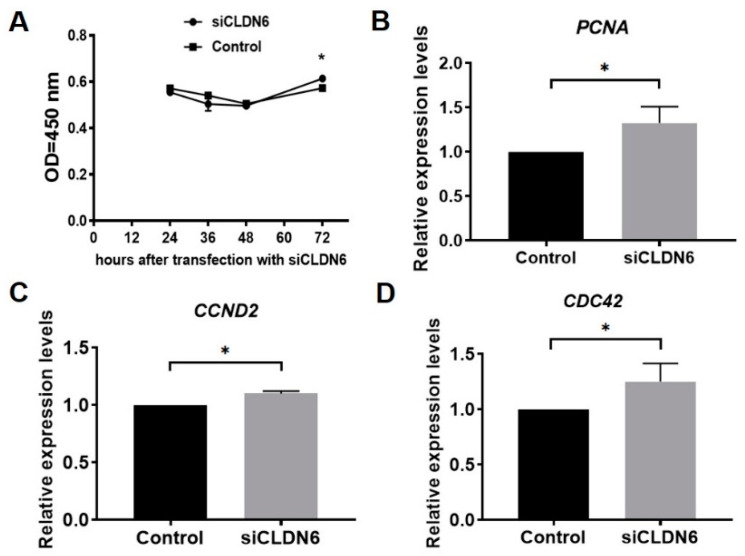
Proliferation analysis of bovine CCs with CLDN6 knockdown. (**A**) After transfection with siCLDN6 or siNC, CCs were treated with CCK-8 and the absorbance at 450 nm was detected. * refers to significant difference (*p* < 0.05). (**B**–**D**) The mRNA levels of *PCNA*, *CCND2* and *CDC42* genes were measured in CCs transfected with siCLDN6 or siNC by RT-qPCR, respectively. Control group was the cells transfected with siNC. * *p* < 0.05.

**Figure 6 ijms-23-13222-f006:**
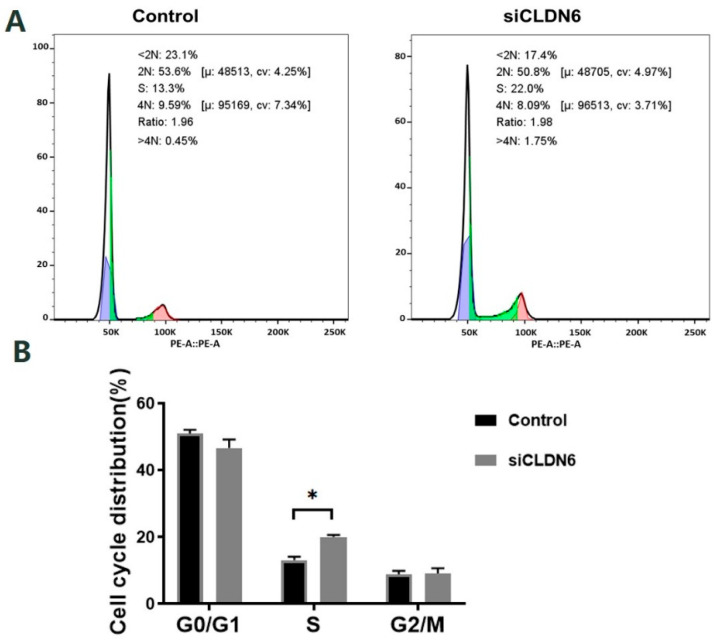
Cell cycle of bovine CCs with CLDN6 knockdown. (**A**) After transfected with siCLDN6 or siNC, CCs were stained with PI and the DNA content of cells was detected by flow cytometry at 488 nm; (**B**) The distribution of CCs at different cell cycle phases. Control group was the cells transfected with siNC. * *p* < 0.05.

**Table 1 ijms-23-13222-t001:** siRNA sequences targeting the bovine *CLDN6* gene.

siRNA	Sequences (5′–3′)	
siRNA-1	Sense Antisense	GCUUCCGCUGGUCUGCAAATT UUUGCAGACCAGCGGAAGCTT
siRNA-2	Sense Antisense	GGAAGGUGACCGCCUUCAUTT AUGAAGGCGGUCACCUUCCTT
siRNA-3	Sense Antisense	GGACCGCCCACACCAUCAUTT AUGAUGGUGUGGGCGGUCCTT
Negative control (NC)	Sense Antisense	UUCUCCGAACGUGUCACGUTT ACGUGACACGUUCGGAGAATT

**Table 2 ijms-23-13222-t002:** Primer sequences for RT-qPCR.

Genes	Accession No.	Primer Sequences (5′–3′)	Product Length (bp)	Annealing Temperature (°C)
*CLDN6*	NM_001205697.1	F: GCUUCCGCUGGUCUGCAAATT	192	59
		R: UUUGCAGACCAGCGGAAGCTT		
*BAX*	NM_173894	F:GGCTGGACATTGGACTTCCTTC	112	61
		R:TGGTCACTGTCTGCCATGTGG		
*BCL-2*	NM_001077486	F:GATGACTTCTCTCGGCGCTA	165	60
		R:GACCCCTCCGAACTCAAAGA		
*CASPASE-3*	NM_001077840	F:TACTTGGGAAGGTGTGAGAAAACTAA	71	59
		R:AACCCGTCTCCCTTTATATTGCT		
*PCNA*	NM_001034494	F:GTCCAGGGCTCCATCTTGAA	123	58
		R:CAAGGAGACATGAGACGAGT		
*CDC42*	NM_001046332	F:GTTGTTGTGGGTGATGGTGC	128	58
		R:TCCCCACCAATCATAACTGT		
*CCND2*	NM_001076372	F:TGACCGCTGAGAAGTTATGC	104	59
		R:CGCCAGGTTCCATTTCAACT		
*GAPDH*	NM-001034034.2	F:GGGTCATCATCTCTGCACCT	177	59
		R:GGTCATAAGTCCCTCCACGA		

## Data Availability

The data used to support the findings of this study are available from the corresponding author upon reasonable request.

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
