# Peer review of "Knockdown of CLAUDIN-6 Inhibited Apoptosis and Induced Proliferation of Bovine Cumulus Cells"

_ijms, 2022, doi:10.3390/ijms232113222_

Round 1

Reviewer 1 Report

The manuscript describes the expression of CLDN6 and its effects on cell proliferation, apoptosis and cell cycle of bovine CCs.

1.      Why did the authors decide to use bovine CCs? Were the cells obtained from healthy or cancer tissue, young or adult?

2.      Why did the authors knockdown CLDN6 in cumulus cells if there is evidence that its expression is higher in cancer cells?

3.      Why did the authors aim their research on the apoptosis of cumulus cells? They should clarify it.

4.      Why did the authors use knockdown instead of a knockout?

5.      How were the bovine CCs separated from the cumulus-oocyte complex? If specific cell markers were used (i.e. phenotype) it needs to be clarified.

6.      How can the authors explain that transfection with siRNA-1 and siRNA-3 wasn’t so efficient?

7.      Did the authors measure apoptosis and cell proliferation after some stimuli as well?

8.      Why did the authors check only caspase-3?

9.      Why the authors didn’t use the same time lapse (24, 48, 36, 72 hours) in all experiments?

10.   Cell proliferation should be measured and confirmed with other methods e.g. clonogenic assay.

11.   What was used as a negative control?

12.   The knockdown effect of CLDN6 should be confirmed on other cells.

13.   Can these results be translated to a clinical setting?

14.   Figure 4B should be changed to be clearer.

15.   The statistical significance marked by a, b, c should be indicated with stars. P-value should also be clarified.

Reviewer 2 Report

Only in the discussion section is a comment made not to exceed in discussing or writing the results of other studies

Round 2

Reviewer 1 Report

The authors answered all questions with valid points.

The statistical significance marked by a, b, c and d should be indicated with stars in all graphs.
